

# SSR marker-based genetic diversity and structure analyses of *Camellia nitidissima* var. *phaeopubisperma* from different populations

Yang-Jiao Xie[1,*], Meng-Xue Su[1,*], Hui Gao[2], Guo-Yue Yan[1], Shuang-Shuang Li[1], Jin-Mei Chen[1], Yan-Yuan Bai[1] and Jia-Gang Deng[3,4,5]

[1] Guangxi University of Chinese Medicine School of Yao Medicine, Nanning, Guangxi, China
[2] Guangxi University of Chinese Medicine School of Nursing, Nanning, Guangxi, China
[3] Guangxi Key Laboratory of Efficacy Study on Chinese Materia Medica, Nanning, Guangxi, China
[4] Collaborative Innovation Center of Research on Functional Ingredients from Agricultural Residues, Nanning, Guangxi, China
[5] Guangxi Key Laboratory of TCM Formulas Theory and Transformation for Damp Diseases, Nanning, Guangxi, China
* These authors contributed equally to this work.

Corresponding authors
Yan-Yuan Bai, 337125802@qq.com
Jia-Gang Deng, dengjg53@126.com

## ABSTRACT

**Background:** *Camellia nitidissima* var. *phaeopubisperma* is a variety in the section *Chrysantha* of the genus *Camellia* of the family Theaceae which is native to Fangchenggang, Guangxi, China. To date, the genetic diversity and structure of this variety remains to be understood.

**Methods:** In the present study, eight simple sequence repeat (SSR) molecular markers previously screened were used to analyze the genetic diversity and structure of *C. nitidissima* var. *phaeopubisperma* natural populations from 14 growing areas in China, so as to determine the influence of environmental changes on genetic variations and provide the basis for introduction and selection of suitable growing sites of that variety.

**Results:** Our results show that, for the eight SSR loci, the observed numbers of alleles per locus (*Na*) and the effective numbers of alleles per locus (*Ne*) were nine and 3.206, respectively on average, and the *Ne* was lower than the *Na* for all loci; the observed heterozygosity (*Ho*) was lower than the expected heterozygosity (*He*). For all the eight loci, the fixation index (*F*) was greater than 0, and the intra-population inbreeding coefficient (*Fis*) for seven loci was positive. Three loci were moderately polymorphic (0.25 < polymorphism information content (PIC) <0.5), and five loci were highly polymorphic (PIC > 0.5); all eight loci had a moderate genetic differentiation level (0.05 < genetic differentiation coefficient (*Fst*) <0.25). As shown by the genetic diversity analysis, the *He* was bigger than the *Ho* for 10 populations, indicating the presence of a certain degree of intra-population inbreeding. The *F* had a negative value for four populations, suggesting that excessive random mating was present within each of them. Results of the analysis of molecular variance show that 19% of the total variation was attributed to among-individuals and 78% of the total variation originated from within-individuals. The adjusted *Fst* (*F'st*) was 0.073, indicative of a moderate level of genetic differentiation among the populations. The
value of gene flow was greater than 1 (7.367), suggesting that genetic differentiation among populations was not caused by genetic drift. Results of the STRUCTURE analysis show that all the samples tested could be clustered into five ancestor groups. Results of the Unweighted Pair Group Method using Arithmetic Averages (UPGMA) clustering analyses show that the 84 plant samples could be divided into three clusters and natural populations from the 14 growing areas could be divided into two clusters. Clustering results of the populations were not affected by geographic distances, and gene flow occurred frequently among the populations, suggesting that the genetic variation among the natural populations of *C. nitidissima* var. *phaeopubisperma* from 14 growing areas was not influenced by environmental changes of these areas but mainly derived from the genetic variation present in pre-introduction populations.

## INTRODUCTION

More than 50 members in the section *Chrysantha* of the genus *Camellia* of the family Theaceae have been discovered all over the world according to documentary records (*Hong, Chen & Lin, 2020*; *Le et al., 2023*). *C. niidissima* C.W.Chi is a type species and the first species to be discovered in the section *Chrysantha* (*Ye & Xu, 1992*). *C. nitidissima* var. *Phaeopubisperma* is a variety of *C. niidissima*, possessing the most abundant resources and being most extensively introduced and cultivated among all golden camellia species and varieties. As the main source of commercial golden camellia, *C. nitidissima* var. *phaeopubisperma* boasts important medicinal, economic, and ornamental value. Its leaves have pharmacological effects such as anti-oxidation, anti-tumor, and lowering blood sugar and blood lipid (*Chen et al., 2021a*; *Wang et al., 2022*; *Chen et al., 2021b*; *Zhang et al., 2020*; *Wan et al., 2011*; *Zhao et al., 2022*; *Zhang, Wu & Qin, 2020*). *C. nitidissima* var. *phaeopubisperma* is native to Fangchenggang, Guangxi, China, and has been introduced to other places of Guangxi, Guangdong, Fujian and other provinces of China. The quality of herbal drugs may alter with environmental conditions. Environmental conditions affect herbal drug quality by directly affecting the synthesis of active ingredients in herbal drugs (*Qu, Chen & Zhao, 2024*; *Tava et al., 2019*), or by causing genetic variations in species (*Shang et al., 2024*; *Cui et al., 2023*), which essentially affects the synthesis of active ingredients in herbal drugs. Assessing quality differences among golden camellia from various regions is essential for advancing the industry. Unfortunately, no information regarding quality differences among golden camellia growing in different habitats has been available so far.

The ability of species to adapt to diverse environments is largely dependent upon their genetic diversity (*Gong et al., 2024*), which plays an important role in sustainability and adaptability in agriculture. For example, a high level of genetic diversity in rice landraces in Yunnan, China contributes to high grain quality, wide adaptability, and strong environmental tolerance (*Cui et al., 2021*); the genetic diversity in different lines of rice in Egypt may be helpful for the selection of drought-tolerant line(s) (*Gaballah et al., 2021*);

the rich genetic diversity of the *Passiflora* species in the karst region of Guizhou, China indicates that these species can potentially be used in breeding programs in a sustainable manner (*Wu et al., 2024a*). Genetic diversity is the cumulative result of genetic variations that can be used to deduce the adaptation mechanism and evolution process of species in different environments (*Colicchio et al., 2021*). Plants growing in different environments may develop their own unique genetic structure and population characters due to the difference in climate, soil, altitude and other ecological factors (*Gong et al., 2024*; *Hoogerheide et al., 2024*). At the same time, anthropic activities such as gathering, planting, and exploitation may also have impacts upon plant populations to varying degrees, which further increases the complexity of their genetic diversity and population differentiation (*Tisthammer et al., 2020*).

Simple sequence repeats (SSRs) are sequence-repeating units ubiquitous in eukaryotic genomes. The number of SSR repeat units is highly variable, leading to polymorphism of loci. SSRs are abundant, highly polymorphic, and codominant (*Li et al., 2024*). Different from the universality of other markers in all species, SSR molecular markers are species-specific, so they are an ideal tool for the analysis of genetic diversity of the same species (*Yan et al., 2024*). SSR markers have been widely used in the analysis of plant genetic relationships and population structure (*Li et al., 2020*; *Cui et al., 2021*; *Bai et al., 2022*; *Li et al., 2022*; *Yan et al., 2022*; *Zhu et al., 2022*; *Cheng et al., 2024*; *Wu et al., 2024a*). The analysis of genetic differences in SSRs between golden camellia from different growing areas is helpful in elucidating the influence of environmental conditions of different growing areas on the quality of golden camellia from a genetic perspective. Therefore, we used SSR markers to analyze the genetic diversity and structure of *C. nitidissima* var. *phaeopubisperma* in this study.

Geographical isolation and environmental heterogeneity influence the pattern of gene flow (*Cui et al., 2021*), contributing to genetic differentiation (*Liu et al., 2023*), and restricted gene flow even drives speciation (*Loretán et al., 2020*). Since the populations of *Camellia nitidissima* var. *phaeopubisperma* we sampled came from different habitats being far away from each other and having different environmental conditions, their interplay *via* seed dispersal by wind, insects, birds or other animals may be limited, so we predicted that they may show significant levels of genetic differentiation. Briefly, we attempted to determine whether environmental changes may have influence on the genetic variation of *Camellia nitidissima* var. *phaeopubisperma*.

## MATERIALS AND METHODS

### Plant materials and DNA extraction

In March 2023, we gathered newly-grown healthy leaves of *C. nitidissima* var. *phaeopubisperma* from 14 golden camellia planting bases in Guangxi and Guangdong, China. All the 14 planting bases, except two located in Hepu County in Guangxi (with a distance of 19 km), have a distance greater than 50 km from each other. All golden camellia individuals from each planting base formed a natural population, and 14 natural populations were included in this study. At least five representative golden camellia individuals were randomly chosen from each natural population at each planting base, and

a total of 84 individuals were sampled (see Table S1 in the Supplemental Materials for sample codes and collection sites). The geographical and climatic parameters of the sampling sites as well as the sample size is shown in Table 1. Soil composition at the 14 sampling sites is presented in Table S2. The climatic and soil conditions vary across these planting bases. All individuals of *C. nitidissima* var. *phaeopubisperma* at the 14 plant bases originated from Fangchenggang, Guanggxi, growing from cuttings taken from wild plants in Fangchenggang for more than 15 years.

The collected leaves were quickly dried with silica gel and stored at 4 °C for DNA extraction and PCR amplification. Total DNA in leaves was extracted (see Appendix 1 in the Supplemental Materials for DNA extraction procedure) using the magnetic bead-based kit for plant tissue DNA extraction from Wuhan Tianyi Huayu Gene Technology Co., Ltd. (Wuhan, China) and stored at −20 °C for PCR test. DNA concentration and quality were determined by NanoDrop 8000 and 1% agarose gel electrophoresis, respectively.

## Primer screening and PCR amplification

The amplification effect and polymorphism of 32 pairs of primers (see Table S3 in the Supplemental Materials) screened in our previous study (*Gao, 2022*) as well as the efficiency with which they evaluated golden camellia diversity were analyzed. The result showed that the evaluation effect on the diversity of golden camellia obtained by the use of only eight pairs of primers (see Table 2) was the same as by using the above 32 pairs of primers. Therefore, the eight pairs of screened primers were used for amplification of DNA isolated from the 84 plant samples. The SSR fluorescent primer system used for PCR amplification had a volume of 10 µL, consisting of 1.0 µL of fluorescent PCR products, 0.5 µL of GeneScan$^{TM}$ 500 LIZ, and 8.5 µl of Hi-Di$^{TM}$ Formamide. The amplification procedure is shown in Table S4 in the Supplemental Materials. Following the PCR test, the amplification product was detected by fluorescence-labeled capillary electrophoresis. See Appendix 2 for specifics about sample preparation for capillary electrophoresis.

## SSR detection and data arrangement

Formamide and relative molecular weight internal standard were mixed thoroughly at a ratio of 250:1 (v/v). A total of 9 µL of the resultant mixture was loaded into the microplate, and 1 µL of PCR product diluted tenfold was added. The ABI 3730xL DNA sequencer was used for capillary electrophoresis, and the original data obtained by it were analyzed by the Fragment (Plant) fragment analysis software in GeneMarker. The position of the internal standard in each lane and the peak position of each sample were compared and analyzed to obtain the size of fragments. The internal standard score for each of test data must be at least 90. During allele calling, the height of a single peak without any accompanying peak(s) within the Panel range should be no less than 250, and the height of the main peak in case of multiple peaks should be at a minimum of 500; the peak(s) must be sharp and bilaterally symmetrical and no inverted peak(s) or nested peaks (peaks of more than two kinds of fluorescence appearing at the same position) were allowed. Raw data in .fsa format were exported from the ABI 3730xL DNA sequencer and imported into the GeneMarker analysis software after sorting and filing by loci detected for genotype data reading, and the

**Table 1 Sampling sites, samples from each site representing a natural population.**

| No. | Sampling sites | Site code | I (°) | II (°) | III (°) | IV (°) | V (°) | VI (mm) | VII (hr) | Sample size |
|---|---|---|---|---|---|---|---|---|---|---|
| 1 | Zhuguang Farm, Shikang, Hepu, Beihai, Guangxi, China | SK | 109.22 | 21.68 | 16.9 | 28.0 | 1.0 | 1,554 | 1,288 | 5 |
| 2 | Longmenjiang, Lianzhou, Hepu, Beihai, Guangxi, China | HP | 109.23 | 21.67 | 16.1 | 40.6 | −0.1 | 1,600 | 1,286 | 5 |
| 3 | Renli, Naliang, Dongxing, Fangchenggang, Guangxi, China | RL | 107.88 | 21.63 | 16.0 | 40.0 | −18.0 | 1,624 | 1,286 | 10 |
| 4 | Chongfeng, Jiangping, Dongxing, Fangchenggang, Guangxi, China | CF | 108.74 | 21.38 | 15.6 | 41.0 | −13.2 | 1,781 | 1,277 | 5 |
| 5 | Shishui, Shapo, Bobai, Yulin, Guangxi, China | BB | 109.88 | 21.83 | 16.4 | 40.8 | −12.7 | 1,477 | 1,275 | 5 |
| 6 | Guigang Agricultural Science and Technology Research Institute, Guangxi, China | GG | 109.55 | 23.12 | 16.3 | 33.0 | −3.0 | 1,536 | 1,277 | 5 |
| 7 | Dongzhuang, Huajiang, Xing'an, Guilin, Guangxi, China | XA | 110.41 | 25.70 | 15.9 | 38.0 | −7.4 | 1,630 | 1,643 | 5 |
| 8 | Xiaocheng, Shuangjiang, Lipu, Guilin, Guangxi, China | LP | 110.40 | 24.57 | 15.0 | 42.7 | −6.8 | 1,500 | 2,113 | 6 |
| 9 | Yimiaotang, Etang, Pinggui, Hezhou, Guangxi, China | HZ | 111.59 | 24.32 | 16.0 | 41.5 | −8.2 | 1,430 | 1,784 | 13 |
| 10 | Tiantang, Changtang, Qingxiu, Guangxi, China | CT | 108.69 | 22.81 | 18.0 | 40.0 | −5.0 | 1,800 | 1,668 | 5 |
| 11 | Gucheng, Mingliang, Shanglin, Nanning, Guangxi, China | SL | 108.58 | 23.37 | 18.5 | 31.0 | 8.0 | 1,756 | 1,741 | 5 |
| 12 | Xialong'an, Fucheng, Wuming, Nanning, Guangxi, China | WM | 108.21 | 23.41 | 18.0 | 40.0 | −5.0 | 1,600 | 1,900 | 5 |
| 13 | Hecun, Laituan, Jiangzhou, Chongzuo, Guangxi, China | CZ | 107.49 | 22.50 | 17.1 | 38.7 | −6.9 | 1,775 | 1,739 | 5 |
| 14 | Wanzhang Golden Camellia Planting Base, Dianbai, Maoming, Guangdong, China | WZ | 111.19 | 21.80 | 21.7 | 39.6 | −2.4 | 1,713 | 1,917 | 5 |

**Note:**

I, longitude; II, latitude; III, annual average temperature; IV, annual maximum temperature; V, annual minimum temperature; VI, annual rainfall; VII, annual average sunshine.

genotype raw data in Excel format and the genotyping peak chart in PDF format were exported by the name of loci.

## Analyses of genetic diversity and differentiation

Based on PCR amplification results, GenAlEx 6.501 was used to estimate various genetic parameters, including observed number of alleles per locus ($Na$), effective number of alleles per locus ($Ne$), Shannon information index ($I$), polymorphism information content ($PIC$), observed heterozygosity ($Ho$), expected heterozygosity ($He$), fixation index ($F$), intra-population inbreeding coefficient ($Fis = (Hs − Hi)/Hs$, where $Hi$ is the number of expected heterozygotes for individuals and $Hs$ is the mean of expected heterozygotes for each population), inter-population inbreeding coefficient ($Fit = (Ht − Hi)/Ht$, where $Ht$ is the overall number of expected heterozygotes), genetic differentiation coefficient ($Fst = (Ht − Hs)/Ht$), and gene flow value ($Nm = 0.25(1 − Fst)/Fst$) (*Wright, 1931*).

## Genetic structure analysis

By referring to the method described in *Wu et al. (2024b)*, STRUCTURE 2.3.4 was used to analyze the genetic structure of 84 plant samples and 14 populations with K = 1~20, 10,000 burn-in cycles (the model reaching a steady state), and 100,000 Markov chain Monte Carlo (MCMC) iterations resulting in a progressive convergence toward more reliable statistical results (*Porras-Hurtado et al., 2013*). Each K value was run 20 times and the optimal ΔK value (which denotes the best clustering result) was obtained. The genetic structure graph was generated according to the optimal K value. Using find.cluster in R 4.4.1 (adegenet),

**Table 2 Characteristics of microsatellite loci developed for *Camellia nitidissima* var. *phaeopubisperma*.**

| Locus | Allele size range (bp) | Number of alleles | Repeat motif | FPr1 (5′-3′) | RPr1 (5′-3′) |
|---|---|---|---|---|---|
| UG000546 | 194–227 | 10 | (AAG)7 | TGTTTGAGCGACGATTCTTG | GAGAAGGGGGAGAATTGGAG |
| UG002254 | 115–153 | 13 | (GAA)8 | GTATGGCGGAGAAGGATTGA | TCCATAAGGTTCGCCTTTGT |
| UG009875 | 250–281 | 11 | (TCT)8 | CCCTCCATATCCTCCCTCTC | GCCCCAACAAACCATTCTTA |
| UG015958 | 206–224 | 7 | (CTT)7 | TCAGTTCATGATCGGGACAA | CAATCTACCCATCCGAAGGA |
| UG039856 | 147–170 | 9 | (CAC)11 | CCCCACCAATTCTCTCAAAA | AAGGGCGTAGAGTTCCGATT |
| UG044988 | 176–206 | 10 | (CAA)14 | AGAATGCCATCCTCCAACAG | ATTGGGTCCATACCAGGTGA |
| UG049344 | 118–140 | 7 | (TGA)8 | CCCCCAAAGTTCATTCTTCA | GAGAAAGCCCATTTGCTCTG |
| UG290536 | 241–256 | 5 | (GAT)7 | GCTATGGCTCTTGGAAGCAC | GGGCTGATGAGGAGTACCAA |

the K-means algorithm was run to generate a graph, in which different K values were the horizontal coordinate and the obtained BIC values were the vertical coordinate; the K value at the first inflection point was selected to determine the number of clusters of the 84 plant samples. The powermarker software was used to estimate the genetic distances between the populations (samples). The Unweighted Pair Group Method using Arithmetic Averages (UPGMA) clustering based on Nei's genetic distances was performed on all samples and clustering charts were generated. GenAlEx 6.501 was used to run principal coordinates analysis (PCoA) on all samples based on Nei's genetic distances. PCoA can reveal the difference between two samples by intuitively comparing the linear distance between the samples on coordinate axes. The shorter the linear distance between two samples, the smaller the difference between them, or vice versa. Here we used this dimensionality reduction technique to map high-dimensional golden camellia data to the low-dimensional coordinate space to clearly show the difference in genetic relationships among golden camellia from different growing areas, so that we could identify the influence of geographic distances on golden camellia genetic relationships.

## RESULTS

### Genetic diversity

The statistics of genetic diversity and heterozygosity for the eight SSR loci are shown in Table 3. By using the eight pairs of primers, 72 alleles ($Na$) were observed in the 84 plant samples, with a range of 5–13 alleles and a mean of nine alleles for all loci. The number of effective alleles ($Ne$) ranged from 1.576 to 6.045 (3.206 on average). For all loci the $Ne$ was lower than the $Na$, indicating the interaction between alleles. The $I$, $Ho$, and $He$ ranged from 0.732 to 2.028 (1.345), from 0.274 to 0.679 (0.482), and from 0.366 to 0.835 (0.611), respectively. The $Ho$ was lower than the $He$ and the $F$ was bigger than 0 in all the eight loci, indicating a departure from the Hardy-Weinberg equilibrium due to homozygous excess and heterozygote deficiency. The $PIC$ ranged from 0.334 to 0.816 (0.579), $0.25 < PIC < 0.5$ for three loci (moderately polymorphic) and $PIC > 0.5$ for five loci (highly polymorphic).

**Table 3 Genetic parameters, F-statistics, and gene flow value of eight SSR loci in *C. nitidissima* var. *phaeopubisperma*.**

| Locus | Na | Ne | I | Ho | He | F | PIC | Fis | Fit | Fst | Nm |
|---|---|---|---|---|---|---|---|---|---|---|---|
| UG000546 | 10 | 4.011 | 1.595 | 0.560 | 0.751 | 0.255 | 0.710 | 0.147 | 0.259 | 0.131 | 1.660 |
| UG002254 | 13 | 6.045 | 2.028 | 0.675 | 0.835 | 0.192 | 0.816 | 0.052 | 0.195 | 0.151 | 1.404 |
| UG009875 | 11 | 5.194 | 1.890 | 0.679 | 0.807 | 0.160 | 0.785 | 0.011 | 0.144 | 0.134 | 1.616 |
| UG015958 | 7 | 1.699 | 0.875 | 0.274 | 0.411 | 0.334 | 0.382 | 0.233 | 0.315 | 0.106 | 2.102 |
| UG039856 | 9 | 2.008 | 1.096 | 0.452 | 0.502 | 0.099 | 0.474 | −0.012 | 0.089 | 0.099 | 2.266 |
| UG044988 | 10 | 2.398 | 1.373 | 0.440 | 0.583 | 0.244 | 0.562 | 0.157 | 0.248 | 0.107 | 2.085 |
| UG049344 | 7 | 2.717 | 1.178 | 0.476 | 0.632 | 0.246 | 0.567 | 0.085 | 0.237 | 0.166 | 1.254 |
| UG290536 | 5 | 1.576 | 0.723 | 0.298 | 0.366 | 0.186 | 0.334 | 0.089 | 0.171 | 0.090 | 2.529 |
| Mean | 9 | 3.206 | 1.345 | 0.482 | 0.611 | 0.215 | 0.579 | 0.095 | 0.207 | 0.123 | 1.865 |

**Note:**

*Na*, observed number of alleles per locus; *Ne*, effective number of alleles per locus; *I*, Shannon information index; *Ho*, observed heterozygosity; *He*, expected heterozygosity; *F*, fixation index; *PIC*, polymorphism information content; *Fis*, intra-population inbreeding coefficient; *Fit*, inter-population inbreeding coefficient; *Fst*, genetic differentiation coefficient; *Nm*, gene flow.

The *Fis* ranged from −0.012 to 0.233 (0.095), its value being positive for seven loci, also indicating deviation from the Hardy-Weinberg equilibrium due to homozygous excess and heterozygote deficiency. The *Fit* had a range of 0.144–0.315 (0.207).

The parameter values representing the genetic diversity of the 14 populations are listed in Table 4. The average value of the *Na*, *Ne*, *I*, *Ho*, and *He* was 3.493, 2.400, 0.910, 0.445, and 0.489, respectively. For populations LP, CF, HZ, WM, CT, SK, SL, EL, WZ, and GG, the *He* was higher than the *Ho*, indicating the presence of a certain degree of inbreeding within the populations. The population HZ had the highest value of *Na*, *I*, and *He*, suggesting that its genetic diversity was higher than that of other populations. The *F* ranged from −0.035 to 0.234 (0.071) for all the populations; populations XA, CZ, HP, and BB had a negative *F* value, indicating the presence of excessive random mating within each of them (they were subjected to inbreeding repression or under the pressure of natural selection).

## Gene flow and genetic differentiation

As shown in Table 3, the *Fst* ranged from 0.090 to 0.166 (0.123), meaning that all loci had a moderate level of genetic differentiation(0.05 < *Fst* < 0.25). The *Nm* ranged from 1.254 to 2.529 (1.865); its value was bigger than 1 for the eight loci, suggesting frequent gene exchange between populations.

Our analysis of molecular variance (AMOVA) result (Table 5) shows that, genetic variations among populations, among individuals, and within individuals accounted for 3%, 19%, and 78%, respectively of the total variation. F statistics for 14 populations were given in Table 6, which shows that the *Fis*, *Fit*, *Fst*, and *F'st* were 0.197, 0.223, 0.033, and 0.073, respectively, suggesting that the populations were at a moderate level of genetic differentiation. The *Nm* was 7.367, greater than 1, suggesting that inter-population genetic differentiation was not a result of genetic drift.

**Table 4 Population genetic diversity analysis results for *Camellia nitidissima* var. *phaeopubisperma*.**

| Population | *Na* | *Ne* | *I* | *Ho* | *He* | *F* |
|---|---|---|---|---|---|---|
| XA | 2.700 | 2.115 | 0.706 | 0.420 | 0.400 | −0.070 |
| LP | 3.900 | 2.718 | 1.022 | 0.467 | 0.531 | 0.092 |
| CF | 3.100 | 2.353 | 0.897 | 0.440 | 0.508 | 0.107 |
| HZ | 5.200 | 2.631 | 1.150 | 0.469 | 0.567 | 0.205 |
| WM | 3.000 | 2.068 | 0.793 | 0.360 | 0.446 | 0.133 |
| CZ | 3.400 | 2.447 | 0.915 | 0.500 | 0.490 | −0.035 |
| CT | 3.600 | 2.620 | 0.992 | 0.480 | 0.534 | 0.116 |
| SK | 3.700 | 2.613 | 1.004 | 0.420 | 0.538 | 0.234 |
| SL | 3.200 | 2.360 | 0.840 | 0.415 | 0.451 | 0.052 |
| EL | 4.200 | 2.569 | 0.947 | 0.380 | 0.471 | 0.169 |
| HP | 3.200 | 2.310 | 0.877 | 0.500 | 0.486 | −0.023 |
| WZ | 3.400 | 2.215 | 0.893 | 0.420 | 0.488 | 0.081 |
| GG | 3.600 | 2.775 | 1.022 | 0.500 | 0.550 | 0.098 |
| BB | 2.700 | 1.802 | 0.679 | 0.460 | 0.390 | −0.171 |
| Mean | 3.493 | 2.400 | 0.910 | 0.445 | 0.489 | 0.071 |

**Note:**
*Na*, observed number of alleles; *Ne*, effective number of alleles; *I*, Shannon information index; *Ho*, observed heterozygosity; *He*, expected heterozygosity; *F*, fixation index.

The gene flow (*Nm*) values and genetic differentiation coefficients (*Fst*) for the 14 populations are presented in Table 7. The *Nm* ranged from 1.759 (HP-BB) to 10.817 (CT-SK), 3.931 on average and bigger than 1 in all cases, indicating frequent gene exchange between those populations, from which it can be deduced that genetic differentiation caused by genetic drift may be prevented among the populations. The *Fst* ranged from 0.023 (SK-CT) to 0.130 (XA-SK), 0.0720 on average, suggesting the presence of moderate genetic differentiation among the populations.

## Genetic structure and relationships

The PCoA result of the 84 plant samples is shown in Fig. 1, from which we know that samples from the same population were distributed in a scattered manner, and genetic distances between most individuals were independent of the population they belonged to, suggesting that the genetic difference among the 84 plant samples was mainly attributed to individuals.

In order to have a deeper understanding of the genetic relationship between the 84 plant samples and between the 14 populations of *C. nitidissima* var. *phaeopubisperma*, Bayesian clustering was performed in the STRUCTURE software to analyze their genetic structure. The change of Delta K with the number of groups (K) is shown in Fig. 2, from which it can be found that Delta K reached the maximum level when K = 5, suggesting that the 84 plant samples from the 14 populations could be categorized into five ancestral groups. Genetic maps of group structure were generated with K = 5 (see Figs. 3A and 3B). In Fig. 3A, all the 84 plant samples from the 14 growing areas inherited genetic properties of the five ancestral groups, and their hereditary materials were contributed by each and all groups.

**Table 5 Analysis of molecular variance results of 14 populations of *Camellia nitidissima* var. *phaeopubisperma*.**

| Source of variation | Degree of freedom | Sum of squares | Mean squares | Estimated variance | Percentage of variation (%) |
|---|---|---|---|---|---|
| Among populations | 13 | 57.297 | 4.407 | 0.094 | 3% |
| Among individuals | 70 | 230.864 | 3.298 | 0.542 | 19% |
| Within individuals | 84 | 186.000 | 2.214 | 2.214 | 78% |
| Total | 167 | 474.161 | | 2.850 | 100% |

**Table 6 F statistics for 14 populations of *Camellia nitidissima* var. *phaeopubisperma*.**

| F-statistics | Value | *P* |
|---|---|---|
| *Fst* | 0.033 | 0.001 |
| *Fis* | 0.197 | 0.001 |
| *Fit* | 0.223 | 0.001 |
| *Fst max* | 0.449 | |
| *F'st* | 0.073 | |
| *Nm* | 7.367 | |

**Note:**
*Fis*, intra-population inbreeding coefficient; *Fit*, inter-population inbreeding coefficient, *Fst*, genetic differentiation coefficient; *Fst max*, maximum *Fst*; *F'st*-adjusted *Fst*; *Nm*, gene flow.

However, the proportion of genes inherited by each sample from the ancestral groups differed, genes from ancestral groups 1–5 mainly inherited by 19, 12, 17, 17, and 19 samples, respectively. In Fig. 3B, the 14 natural populations inherited genes from the five ancestral groups to varying degrees, but the proportion of genes inherited by each of these 14 natural populations from the five ancestral groups also differed. Populations BB, CZ, HP and WM inherited more genes from group 1 (48%, 65%, 55%, and 63%, respectively) than from other groups; population CT inherited more genes from group 3 (47%); population XA inherited more genes from group 1 and group 3 (42% and 44%); populations CF and LP inherited more gene from group 4 (50% and 61%); and population RL inherited more genes from group 5 (52%) (refer to Table S5 in the Supplemental Materials for more information).

K-means clustering was conducted on amplified fragment length data of SSR primers for 84 plant samples. Bayesian information criterion (BIC) values were obtained *via* choosing different K values (the number of clusters). The value of K at the first inflection point was chosen as the optimal number of clusters, and it seemed most appropriate to divide the 84 samples into three clusters (see Fig. 4). The result of UPGMA clustering is shown in Fig. 5, which exhibits that cluster 1 was comprised of 37 samples, cluster 2 consisted of 30 samples, and cluster 3 contained 17 samples (see Table S6 in the Supplemental Materials for specific samples contained in each cluster). Samples from populations CF, HF, HP, and XA were grouped as a cluster, samples from populations BB, CT, LP, RL, SL, and WM fell into two different clusters, and samples from the remaining four populations scattered in the three clusters.

**Table 7 Inter-population gene flow values and genetic differentiation coefficients of 14 populations of *Camellia nitidissima* var. *phaeopubisperma*.**

|     | XA | LP | CF | HZ | WM | CZ | CT | SK | SL | EL | HP | WZ | GG | BB |
|-----|-----|-----|-----|-----|-----|-----|-----|-----|-----|-----|-----|-----|-----|-----|
| XA  |     | 3.137 | 3.082 | 3.987 | 3.435 | 1.997 | 2.628 | 2.433 | 1.677 | 2.620 | 1.759 | 3.895 | 2.631 | 3.124 |
| LP  | *0.074* |     | 4.514 | 4.583 | 4.703 | 4.538 | 5.291 | 5.758 | 4.049 | 8.028 | 4.727 | 2.825 | 5.169 | 2.992 |
| CF  | *0.075* | *0.052* |     | 8.478 | 4.534 | 3.591 | 4.103 | 3.121 | 3.581 | 3.689 | 2.678 | 3.961 | 3.875 | 3.693 |
| HZ  | *0.059* | *0.052* | *0.029* |     | 5.269 | 4.241 | 5.012 | 4.225 | 3.116 | 3.787 | 2.966 | 5.238 | 3.590 | 5.819 |
| WM  | *0.068* | *0.050* | *0.052* | *0.045* |     | 2.643 | 5.284 | 3.730 | 2.831 | 4.037 | 2.457 | 2.817 | 3.281 | 3.364 |
| CZ  | *0.111* | *0.052* | *0.065* | *0.056* | *0.086* |     | 5.122 | 5.547 | 2.726 | 3.459 | 2.522 | 2.601 | 3.185 | 2.689 |
| CT  | *0.087* | *0.045* | *0.057* | *0.048* | *0.045* | *0.047* |     | 10.817 | 3.771 | 4.304 | 2.922 | 2.840 | 4.808 | 2.280 |
| SK  | *0.093* | *0.042* | *0.074* | *0.056* | *0.063* | *0.043* | *0.023* |     | 2.506 | 3.979 | 3.218 | 2.563 | 4.046 | 2.320 |
| SL  | *0.130* | *0.058* | *0.065* | *0.074* | *0.081* | *0.084* | *0.062* | *0.091* |     | 5.099 | 3.323 | 1.922 | 3.365 | 1.790 |
| EL  | *0.087* | *0.030* | *0.064* | *0.062* | *0.058* | *0.067* | *0.055* | *0.059* | *0.047* |     | 4.380 | 2.151 | 4.197 | 2.269 |
| HP  | *0.124* | *0.050* | *0.085* | *0.078* | *0.092* | *0.090* | *0.079* | *0.072* | *0.070* | *0.054* |     | 1.687 | 2.440 | 1.756 |
| WZ  | *0.060* | *0.081* | *0.059* | *0.046* | *0.082* | *0.088* | *0.081* | *0.089* | *0.115* | *0.104* | *0.129* |     | 2.472 | 3.769 |
| GG  | *0.087* | *0.046* | *0.061* | *0.065* | *0.071* | *0.073* | *0.049* | *0.058* | *0.069* | *0.056* | *0.093* | *0.092* |     | 2.038 |
| BB  | *0.074* | *0.077* | *0.063* | *0.041* | *0.069* | *0.085* | *0.099* | *0.097* | *0.123* | *0.099* | *0.125* | *0.062* | *0.109* |     |

**Note:**
Figures in bold are gene flow (Nm) values, for example, 3.137 in row 2 and column 3 is the gene flow between populations XA and LP. Italic figures denote genetic differentiation coefficients, for example, the genetic differentiation coefficient between populations XA and LP is 0.074.

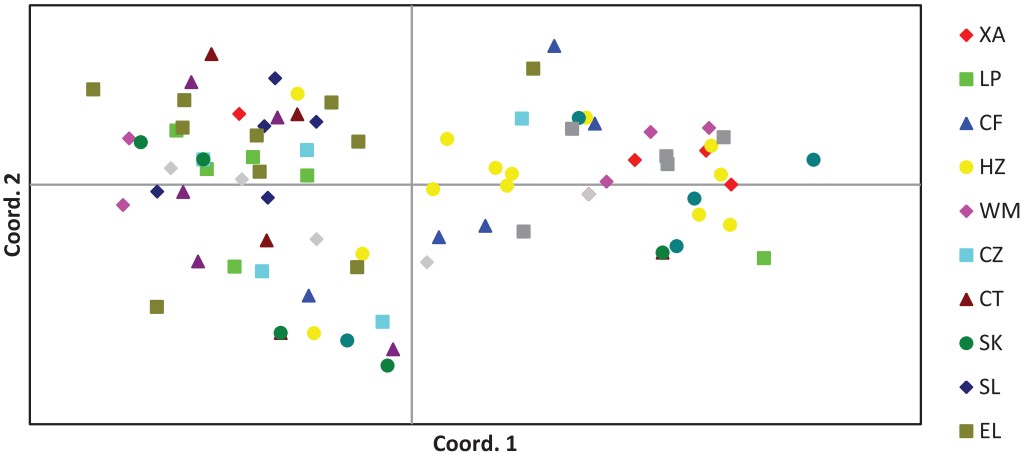

**Figure 1 Principal coordinates analysis result of 84 samples of *Camellia nitidissima* var. *phaeopubisperma*.** The abbreviations on the right stand for populations.

A heat map of genetic distances between the 14 populations is shown in Fig. 6 (see Table S7 in the Supplemental Materials for specific values). The maximum and minimum genetic distances were 0.3601 (WM-SL) and 0.0739 (LP-HZ), respectively. The UPGMA clustering analysis based on Nei's genetic distances shows that the 14 populations could be divided into two clusters (see Fig. 7): one cluster consisted of populations CF, HZ, LP, RL, WZ, SK, and SL, and the other cluster comprised populations GG, BB, CT, HP, XA, CZ,

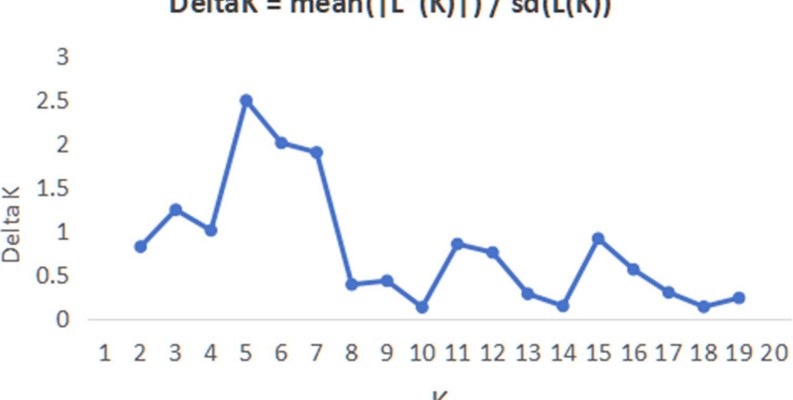

**Figure 2 Graphic plotted with delta K *vs* K.**

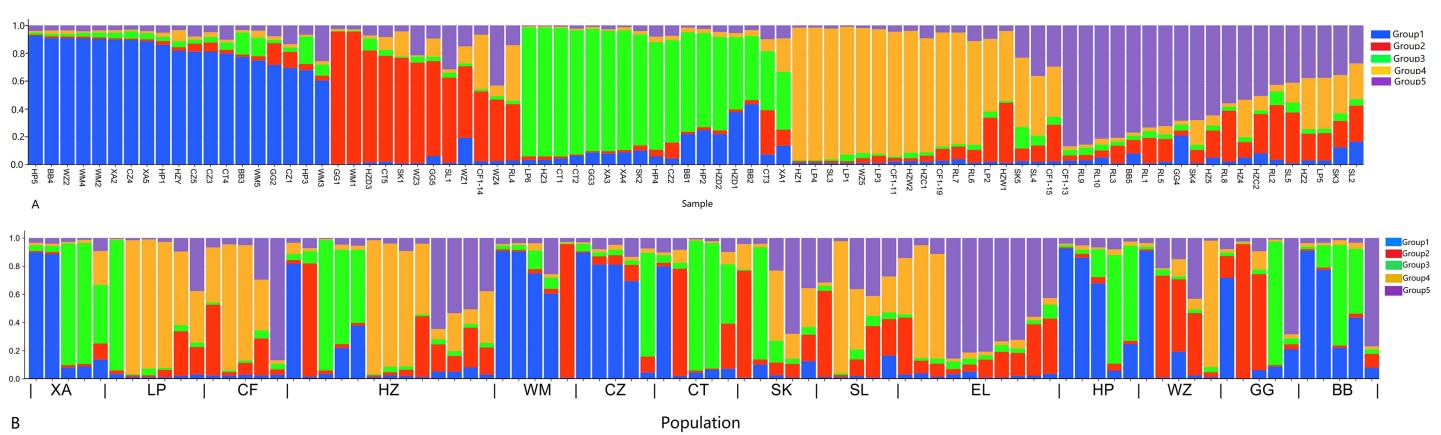

**Figure 3 Structure analysis results of *Camellia nitidissima* var. *phaeopubisperma*.** (A) Assigned proportion of each sample in the groups when K = 5. (B) Assigned proportion of each population in the groups when K = 5. The abbreviations stand for populations, and the number at the end of them denotes the serial number of samples: BB, population from Shishui, Shapo, Bobai, Yulin, Guangxi, China; CF, Population from Chongfeng, Jiangping, Dongxing, Fangchenggang, Guangxi, China; CT, Tiantang, Changtang, Qingxiu, Guangxi, China; CZ, Hecun, Laituan, Jiangzhou, Chongzuo, Guangxi, China; GG, Guigang Agricultural Science and Technology Research Institute, Guangxi, China; HP, Longmenjiang, Lianzhou, Hepu, Beihai, Guangxi, China; HZ, Yimiaotang, Etang, Pinggui, Hezhou, Guangxi, China; LP, Xiaocheng, Shuangjiang, Lipu, Guilin, Guangxi, China; RL, Renli, Naliang, Dongxing, Fangchenggang, Guangxi, China; SK, Zhuguang Farm, Shikang, Hepu, Beihai, Guangxi, China; SL, Gucheng, Mingliang, Shanglin, Nanning, Guangxi, China; WM, Xialong'an, Fucheng, Wuming, Nanning, Guangxi, Chin; WZ, Wanzhang Golden Camellia Planting Base, Dianbai, Maoming, Guangdong, China; XA, Dongzhuang, Huajiang, Xing'an, Guilin, Guangxi, China.

and WM. Populations with a short geographic distance did not necessarily fall into the same cluster, suggesting that the grouping was not dependent of geographic distances.

## DISCUSSION

### Analysis of genetic diversity of SSR loci in *Camellia nitidissima* var. *phaeopubisperma*

Constituting an important part of biodiversity, genetic diversity is the prerequisite for the survival, adaptation, development, and evolution of species, and understanding the genetic diversity of a species not only provides the scientific basis for its conservation and

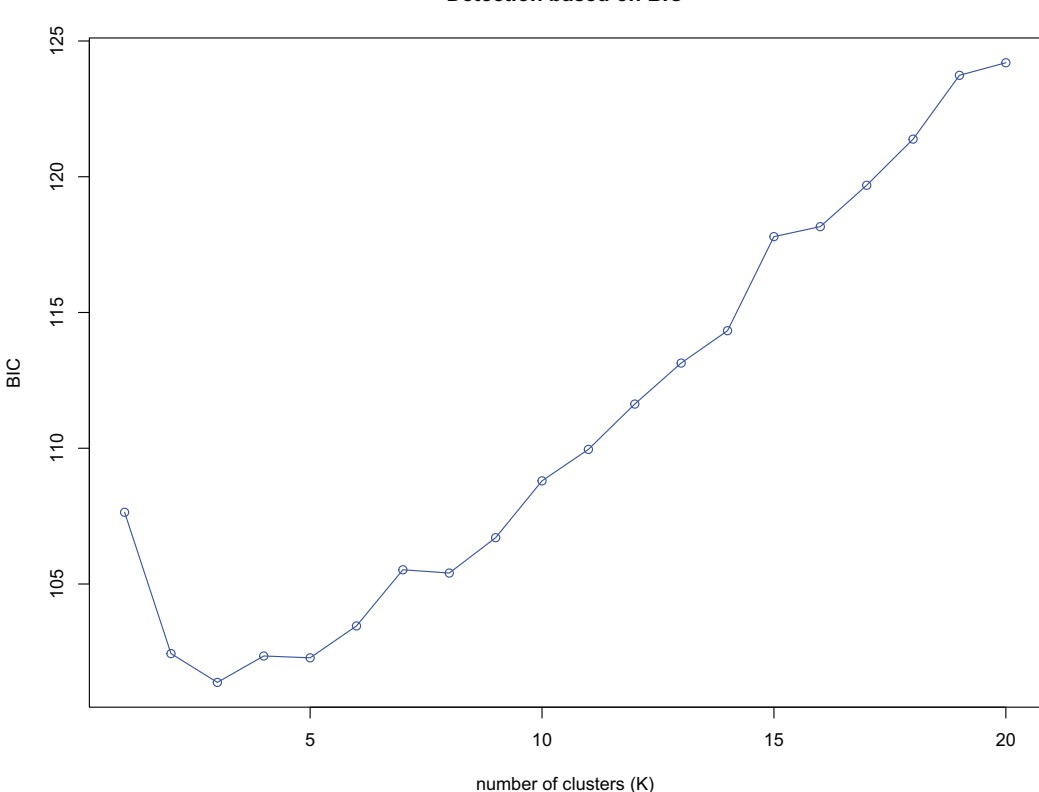

**Detection based on BIC**

**Figure 4 Number of clusters (K) detection based on Bayesian information criterion (BIC).**

utilization, but is of great importance to genetic improvement and innovation of its germplasm resources (*Li et al., 2024*). *C. nitidissima* var. *phaeopubisperma* is the most widely used variety in the section *Chrysantha* and has been introduced to different areas with different environmental conditions. However, no previous report has been made on the status of genetic differentiation of this variety after introduction and cultivation. In this study, we performed SSR genetic diversity analyses on 84 samples of *C. nitidissima* var. *phaeopubisperma* from 14 locations in Guangxi and Guangdong, China, and found that differences in the *Na*, *Ne*, *PIC*, *I*, *Ho*, and *He* were present among the eight screened SSR loci, suggesting that there was a large difference in genetic variations among SSR loci in *C. nitidissima* var. *phaeopubisperma*. This discrepancy is of great significance to germplasm resource evaluation of *C. nitidissima* var. *Phaeopubisperma* since it holds great potential to obtain SSR loci for efficient appraisal of genetic diversity. Five out of eight SSR loci were highly polymorphic (*PIC* > 0.5), and three loci were moderately polymorphic (0.2 < *PIC* < 0.5). According to *Purvis & Franklin (2005)*, loci with PIC > 0.7 are best genetic markers. In our study, the *PIC* was higher than 0.7 for three out of eight SSR loci, meaning that it was possible to choose SSR loci for efficient investigation on genetic diversity of *C. nitidissima* var. *phaeopubisperma* from the tested loci. The *He* ranged from 0.366 to 0.835, indicating substantial genetic diversity among the tested samples (*Wu et al., 2024a*). We also found that the *Ho* was lower than the *He* for all the eight loci, indicating

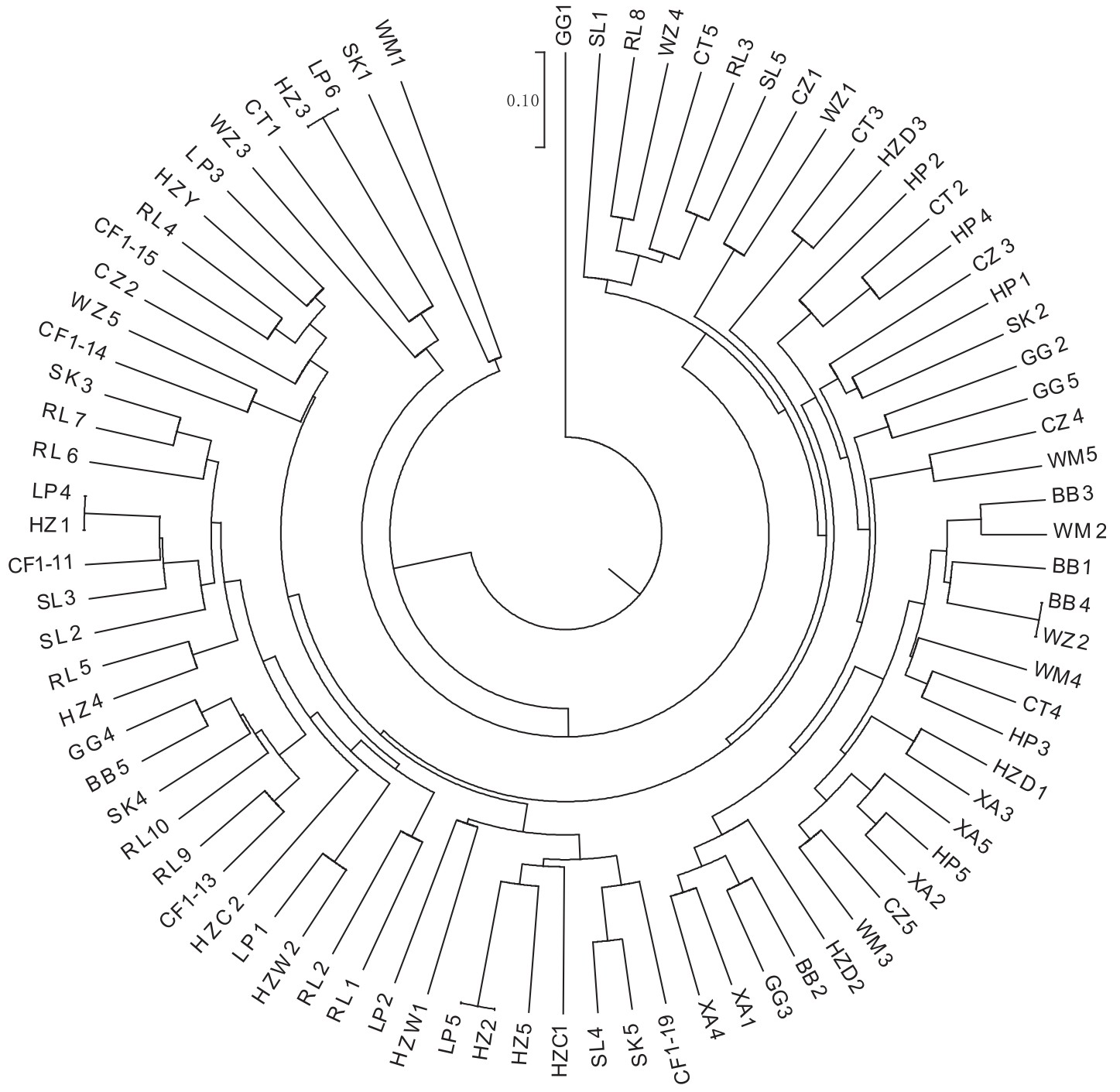

**Figure 5 UPGMA clustering result of 84 samples of *Camellia nitidissima* var. *phaeopubisperma*.** The abbreviations stand for populations, and the number at the end of them denotes the serial number of samples (the same as in Fig. 3).

deviation from Hardy-Weingerg equilibrium in most loci of the progeny populations, possibly due to the influence of non-random factors on parental populations in the process of gene transfer from generation to generation (*Wu et al., 2024b*). Moreover, the genetic differentiation coefficients were higher than 0.05 and lower than 0.25, suggesting a
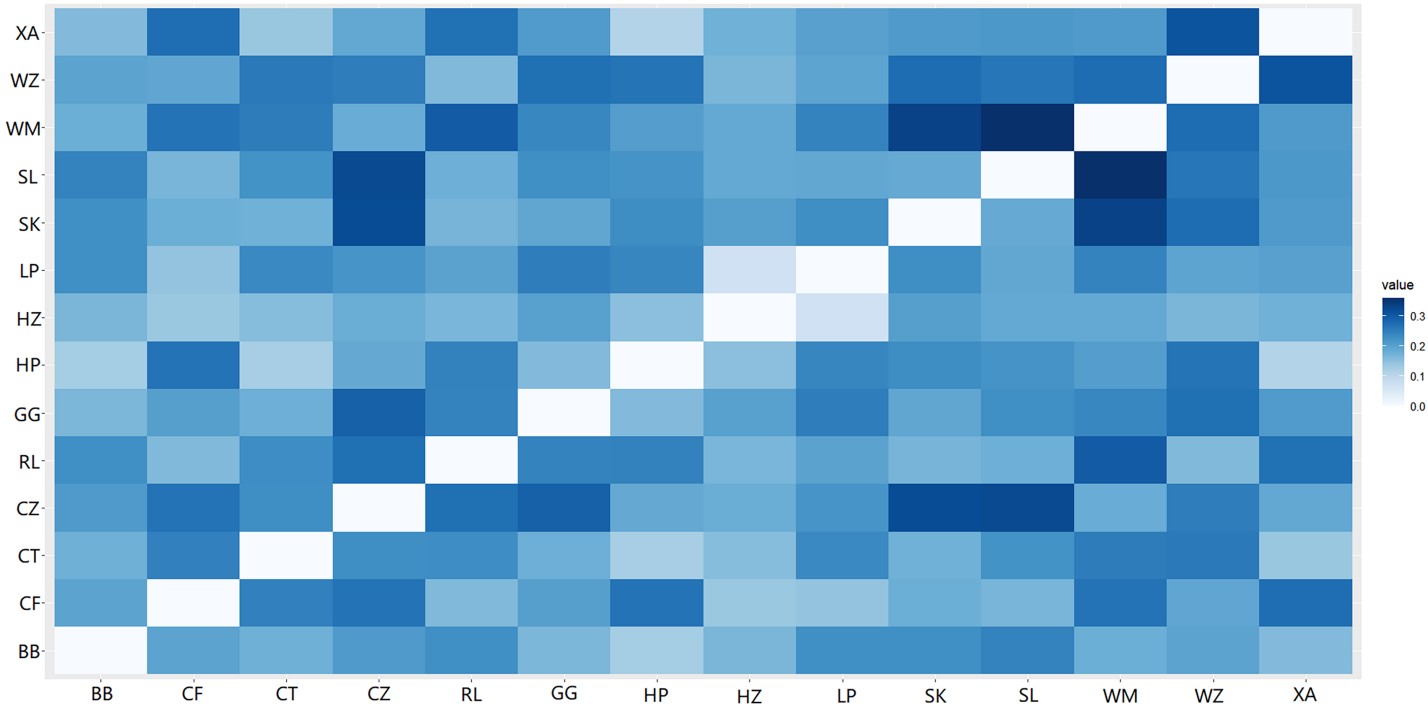

**Figure 6** Genetic distances among populations of *Camellia nitidissima* var. *phaeopubisperma*. The abbreviations stand for populations (the same as in Fig. 3).

moderate level of differentiation (*Stebbins & Wright, 1977*); the gene flow values were greater than 1, suggesting frequent gene exchange among populations and maintenance of current genetic structure. The results of both of these parameters indicated the presence of a moderate level of subdivision of populations (*Wright, 1965*). The populations we studied were characterized by low heterozygosity. Low heterozygosity across a species increases its vulnerability to extinction (*Kramer & Havens, 2009*; *Oakley & Winn, 2012*; *Poudel et al., 2014*) and goes against its sustainable development; however, a low level of heterozygosity may favor the selection and breeding of high-quality germplasms (*Wu et al., 2024b*).

### Analysis of genetic diversity of populations of *Camellia nitidissima* var. *phaeopubisperma*

Previous studies found that no significant correlation exists between the genetic diversity and population size of *C. nitidissima* (*Chen et al., 2022*; *Wei et al., 2008*). There is a high degree of differentiation among populations of this species. Compared to natural populations, *ex-situ* populations derived from transplant of wild seedlings show more genetic variations (*Wei et al., 2008*; *Zhu et al., 2023*). However, the genetic diversity of populations originating from seeds or cuttings from a few high-quality trees is low (*Zhu et al., 2023*). The 14 populations in our study had a moderate level of genetic differentiation, consistent with the result of *Chen et al. (2022)*, but lower than the result of *Wei et al. (2008)*, likely due to the fact that the populations in our study had a higher level of gene flow than those in *Wei et al. (2008)*. Our result validated the conclusion of *Zhu et al. (2023)*. Both *Li et al. (2020)* and *Tang et al. (2006)* stated that genetic distances of

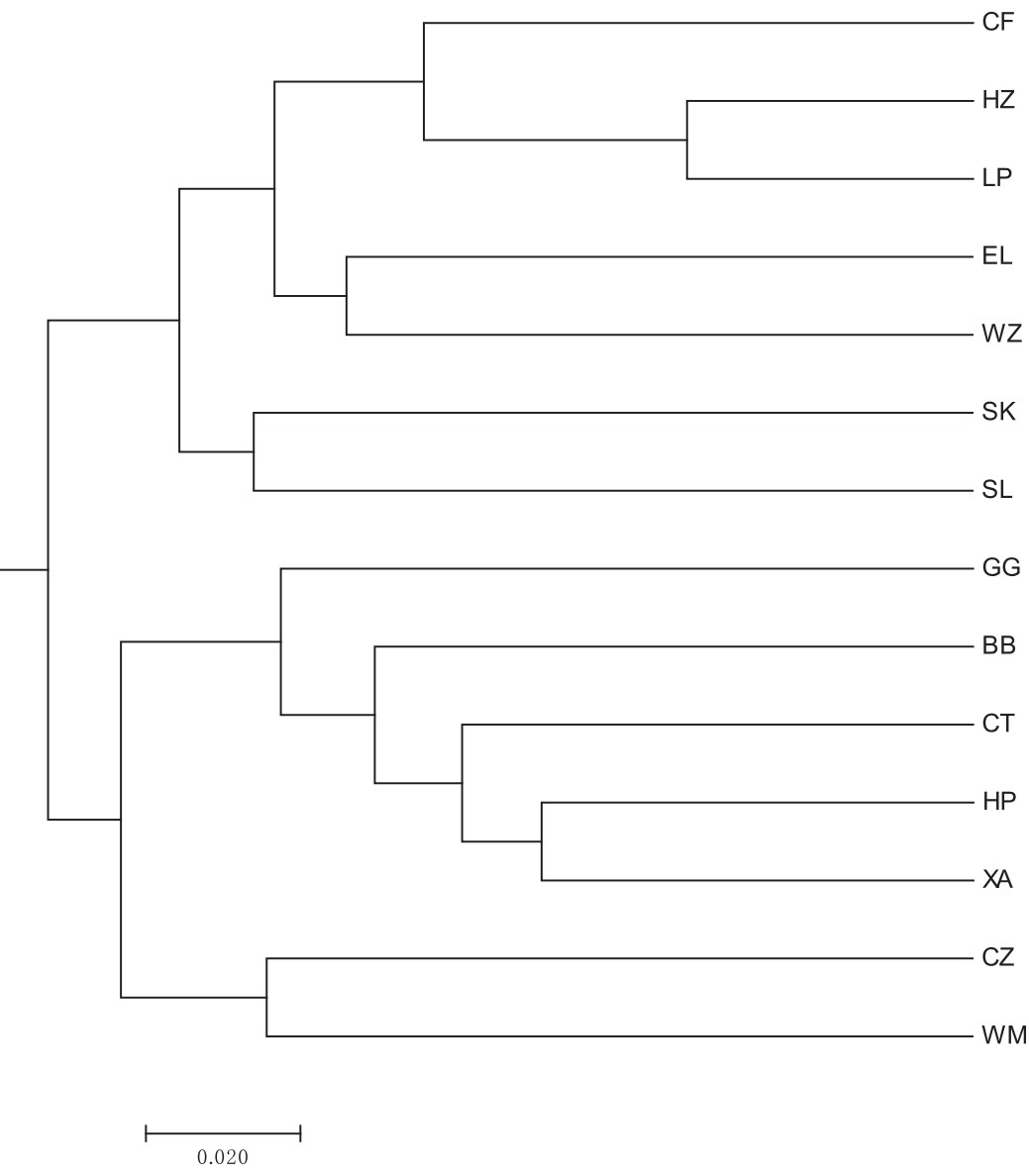

**Figure 7 UPGMA clustering result of 14 populations of *Camellia nitidissima* var. *phaeopubisperma*.** The abbreviations stand for populations (the same as in Fig. 3).

golden camellia are significantly correlated with their geographic distances. In the present study, Nei's genetic distances-based UPGMA clustering analysis performed on 14 natural populations of *C. nitidissima* var. *phaeopubisperma* revealed that these populations could be divided into two clusters. One cluster was composed of populations CF, HZ, LP, RL, WZ, SK, and SL, and the other cluster consisted of populations GG, BB, CT, HP, XA, CZ, and WM (see Fig. 7). It is clear that populations with a short geographic distance did not necessarily fall into the same cluster, suggesting that population clustering was independent of geographic distances. This result was in discord with those of *Li et al. (2020)* and *Tang et al. (2006)*. *Li et al. (2020)* sampled from four sites in Fangchenggang,

Guangxi, China and *Tang et al. (2006)* sampled from six sites in Guangxi, China (four in Fangchenggang, and two in Fusui, Nanning, Guangxi), whereas our study had a higher number of sampling sites (14 in total) and covered more environmental conditions, being more representative of the current genetic status of *Camellia nitidissima* var. *Phaeopubisperma*. Although gene exchanges existed among different populations, there are also genetic subdivision in these populations, this together with the difference in sample size may be the cause of the discrepancy regarding whether there is any correlation between genetic distances and geographic distances. The inter-population gene flow value was 7.367 in our study, bigger than 1, suggesting that frequent gene exchange was present among the populations, inconsistent with the result of *C. tunghinensis* chang in *Tang et al. (2020)*, in which outcrossing mainly occurs between individuals in sub-populations, and there are low levels of gene flow and differentiation between sub-populations. As shown by our AMOVA result (see Table 5), the total genetic variation of *C. nitidissima* var. *phaeopubisperma* was mostly attributed to within-individuals (*i.e.*, the individuals themselves). The low level of inter-population genetic variation in our study may be due to frequent gene exchange among populations. We also conducted STRUCTURE analysis to further explore the genetic relationships among the 14 populations, and the result shows that the 84 samples from the 14 populations were derived from five ancestor groups. Due to the impact from gene flow, the 14 populations inherited genes from the five groups to varying degrees.

## Influences of species and material resources of golden camellia on results of genetic diversity

*C. niidissima* is rare, and the specimens of *C. niidissima* and *C. chrysantha* (Hu) Tuyama used in many documents may be *C. nitidissima* var. *phaeopubisperma*; for example, the Latin name of golden camellia novel food was *C. chrysantha* (Hu) Tuyama as provided in the Announcement of the Ministry of Health of the People's Republic of China (*The Ministry of Health of the People's Republic of China, 2010*), however, after tracing the origin of the samples submitted for inspection to the species level, it was confirmed that these samples were actually *C. nitidissima* var. *phaeopubisperma*. In the present study, the populations growing in different areas originated from seedlings growing from branch cuttings of wild plants identified to be *C. nitidissima* var. *phaeopubisperma*. By contrast, natural populations of *C. nitidissima* were studied in *Wei et al. (2008)*, *Li et al. (2020)*, and *Tang et al. (2006)*, and *C. tunghinensis* chang was studied in *Tang et al. (2020)*. The research materials are different between our study and these previous studies. This difference may lead to different results of genetic diversity, but in essence the discrepancy is due to different levels of gene flow as described above.

Our study has its limitations. For example, the use of transcriptomics alone faces challenges as biological organism might develop adaptive responses which finally can not be explained only from transcriptomics analysis. The integration of transcriptomics and metabolomics might be helpful for further explanation (*Bueno & Lopes, 2020*; *Cadena-Zamudio et al., 2022*; *Fu et al., 2022*).

## CONCLUSIONS

Altogether, the populations investigated in our study were recently at a relatively stable state since they showed population structure, pronounced genetic subdivision, and gene flow. The presence of sub-populations in the populations may be the cause of failure of our results in supporting the correlation between genetic distance and geographic distance. Since gene flow among populations of *C. nitidissima* var. *phaeopubisperma* guarantees stable genetic variations, conservation efforts should focus on preserving existing genetic pools rather than emphasizing geographic isolation or micro-habitat-specific preservation.

## ACKNOWLEDGEMENTS

We thank the Guangxi Scientific Research Center of Traditional Chinese Medicine for providing the experimental site. Our thanks also go to the editor and reviewers for their valuable suggestions.

### Funding

This work was supported by the Guangxi Natural Science Foundation of China (No. 2020GXNSFDA297029), Guangxi University of Chinese Medicine Doctoral Start-up Fund Project (No. 2022BS016), 2020 Operating Subsidy Project of Guangxi Key laboratory of Efficacy study on Chinese Materia Medica (No. 20-065-38), Innovation Project of Guangxi Graduate Education (No. YCSW2024431), Guangxi Major Science and Technology Project (No. Guike AA18118049), and Guangxi University of Chinese Medicine 'Guipai Traditional Chinese Medicine Inheritance and Innovation Team' (2022A005). The funders had no role in study design, data collection and analysis, decision to publish, or preparation of the manuscript.

### Grant Disclosures

The following grant information was disclosed by the authors:
Guangxi Natural Science Foundation of China: 2020GXNSFDA297029.
Guangxi University of Chinese Medicine Doctoral Start-up Fund Project: 2022BS016.
2020 Operating Subsidy Project of Guangxi Key laboratory of Efficacy study on Chinese Materia Medica: 20-065-38.
Innovation Project of Guangxi Graduate Education: YCSW2024431.
Guangxi Major Science and Technology Project: Guike AA18118049.
Guangxi University of Chinese Medicine 'Guipai Traditional Chinese Medicine Inheritance and Innovation Team': 2022A005.

### Competing Interests

The authors declare that they have no competing interests.

## Author Contributions

- Yang-Jiao Xie conceived and designed the experiments, prepared figures and/or tables, authored or reviewed drafts of the article, and approved the final draft.
- Meng-Xue Su conceived and designed the experiments, prepared figures and/or tables, authored or reviewed drafts of the article, and approved the final draft.
- Hui Gao conceived and designed the experiments, analyzed the data, prepared figures and/or tables, authored or reviewed drafts of the article, and approved the final draft.
- Guo-Yue Yan performed the experiments, analyzed the data, prepared figures and/or tables, authored or reviewed drafts of the article, and approved the final draft.
- Shuang-Shuang Li performed the experiments, prepared figures and/or tables, authored or reviewed drafts of the article, and approved the final draft.
- Jin-Mei Chen performed the experiments, prepared figures and/or tables, and approved the final draft.
- Yan-Yuan Bai conceived and designed the experiments, authored or reviewed drafts of the article, and approved the final draft.
- Jia-Gang Deng conceived and designed the experiments, authored or reviewed drafts of the article, and approved the final draft.

## Data Availability

The raw data are available in the Supplemental Files.

## Supplemental Information

Supplemental information for this article can be found online at http://dx.doi.org/10.7717/peerj.18845#supplemental-information.

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
