# Peer review of "SSR marker-based genetic diversity and structure analyses of Camellia nitidissima var. phaeopubisperma from different populations"

_PeerJ, doi:10.7717/peerj.18845_

## Round 0.1 · original submission · Major Revisions

Dear Authors
The manuscript cannot be accepted for publication in its current form. It needs a major revision before publication. The authors are invited to revise the paper considering all the suggestions made by the reviewers. Please note that the requested changes are required for publication.
With Thanks

Reviewer 1 ·

Basic reporting

The manuscript titled "SSR marker-based genetic diversity and structure analyses of Camellia nitidissima var. phaeopubisperma from different populations" provides an in-depth exploration of the genetic diversity and population structure of C. nitidissima var. phaeopubisperma, a species of significant ecological and economic importance. Using SSR markers, the study examines genetic variation across populations from multiple locations in China, contributing valuable insights into the conservation and potential breeding applications for this rare species. While the research is timely and well-conceived, addressing an important gap in our understanding of genetic diversity within this species, there are notable areas where the manuscript's clarity, scientific rigor, and technical precision could be enhanced. While the manuscript attempts to use technical language, it often falls into vague wording or overly general phrases.
Key words: Revise key words, they must be 5-7 and unique than the terminologies used in the title of the manuscript.
In introduction the inclusion of more recent and relevant studies on the genetic adaptation mechanisms in plants under varying environmental conditions would help to establish the study’s significance more convincingly. It could be strengthened by discussing how genetic diversity in commercial crops influences sustainability and adaptability in agriculture. The study’s aim to assess the genetic diversity of C. nitidissima var. phaeopubisperma across different geographical regions using SSR markers is reasonable. However, the novelty of using SSRs in this context is not well justified. SSR markers are widely used; thus, the introduction should better explain why this method is optimal for this study, considering its limitations and strengths relative to other molecular markers. The hypothesis about genetic distance correlating with geographic distance is stated somewhat superficially. Elaborate on the ecological or evolutionary rationale for expecting genetic variation across the geographic areas studied. The language needs refinement for conciseness and scientific accuracy. Phrases such as "possessing the most resources" or "important manifestation of genetic diversity" should be clarified or replaced with more precise terms. Moreover, redundancies, such as repeated mentions of the same references, disrupt readability. While the references cover relevant genetic diversity studies, they rely heavily on older citations and overlook recent advancements in genetic analysis of stress-tolerant plant species. Including recent studies on environmental genetics, especially those applying SSR markers in similar species or environmental contexts.
Material and methods: More information is needed on the criteria for selecting the 14 planting bases and how representative these are of the environmental variation in the broader region. Details on environmental conditions at each sampling site (e.g., soil composition, climate parameters) would strengthen the study's context, particularly given the introduction’s focus on environmental impact on genetic variability. While the use of a commercial kit for DNA extraction is mentioned, the protocol specifics are lacking. Given that DNA extraction can influence yield and quality, a brief description of the extraction procedure and any modifications made could improve reproducibility. Critical PCR parameters (such as annealing temperatures, and buffer conditions) are absent in the section. These factors significantly impact amplification success and data quality, and thus, need to be specified or included in the supplementary materials Table S3. Mention why 8 primers were chosen based on prior studies? The sample preparation for capillary electrophoresis lacks specifics. Details on the loading volume, electrophoresis conditions, and other critical settings are necessary to ensure reproducibility. Additionally, the criteria for quality control during fragment analysis, such as thresholds for peak calling or error correction, are not provided. Please provide to validate the SSR data accuracy. The description of genetic parameters and statistical tools is brief and requires clarification, especially for complex indices such as Fst, Fit, and Fis. The equations or reasoning behind these measurements should be included, or references should be provided for more specific methodologies. In the genetic structure analysis, explaining the rationale for choosing specific parameters (e.g., 10,000 burn-in cycles and 100,000 MCMC iterations) would make the section more transparent. Additionally, software versions and settings (like convergence criteria in STRUCTURE) should be fully detailed. Including more information on the statistical software (R, GenAIex, etc.) used for data analysis and any additional libraries or packages would be beneficial. It would also be helpful to explain why specific methods like PCoA were chosen for visualizing genetic distances.
The "Results" section is comprehensive, but it could benefit from revisions to improve clarity, coherence, and alignment. Some parts of the section contain redundant or overly detailed explanations, especially of standard methodologies (e.g., AMOVA and PCoA methods). This detracts from the focus on results and could be streamlined for clarity. Use of vague phrases, such as "suggesting the presence of homozygous excess and heterozygote deficiency," should be concise and technically precise. These could be replaced with straightforward descriptions, e.g., “indicating a deviation from Hardy-Weinberg equilibrium due to….” Statistical values, like mean allele counts or gene flow rates, are presented with minimal interpretation. Terms like "high diversity" for heterozygosity values could be supplemented with comparative values or references to increase rigor. Results are scattered and lack a cohesive flow. Consider grouping results into broader thematic subsections e.g., “Population Genetic Structure,” “Gene Flow and Differentiation”. Overviews at the start or end of each subsection could synthesize key results, which helps in emphasizing the significance of findings and maintaining focus. Avoid using informal phrases like “middle-level genetic differentiation,” as these lack specificity. Consider standard terms like “moderate genetic differentiation” or quantifiable metrics (e.g., Fst thresholds). The usage of software like STRUCTURE or UPGMA is well-noted, but descriptions lack specifics about model assumptions or parameter choices.
In Discussion, the text is heavily reliant on citations to establish common concepts, such as the general importance of genetic diversity, which are widely understood. Instead, the authors should focus on novel interpretations or findings unique to this study. The discussion tends to reiterate results rather than delve into their implications. The authors should emphasize interpretation, relevance, and future directions rather than simply restate findings. The authors mention that results differ from past studies due to variations in sample origin, environment, and methodology. However, they fail to provide a critical analysis of how these factors influence the observed genetic patterns, missing an opportunity to provide deeper insight into the causes of these discrepancies.
The conclusion opens by negating the hypothesis of a positive association between genetic and geographic distance, but it does not clarify the rationale or significance of this finding within the broader context of population genetics or the species’ evolutionary history. The claim that genetic variation was "independent upon changes in environmental conditions" would be strengthened by briefly discussing why this is significant. Is this stability likely due to gene flow among populations, or are there inherent genetic factors that contribute to limited environmental influence? Please provide practical implications for conservation strategies. For instance, if genetic variation is stable across environmental gradients, then conservation efforts might focus on preserving existing genetic pools rather than emphasizing geographic isolation or micro-habitat-specific preservation.
Please provide the full form of abbreviation used in the Figure 3, 5, 6 and 7 in its foot notes. Remember that figures must be self-explanatory.
Specific comments:
Lines 52-53: "Primarily the part of it used as a medicine is its leaves, which have pharmacological effects such as anti-oxidation, anti-tumor, and lowering blood sugar and blood lipid." Correct the expression.
Line 57: "The quality of herbal drugs may change with environmental conditions of plant habitats." Change to "The quality of herbal drugs may vary with environmental conditions."
Lines 58-60: "on one hand, changes in environmental conditions can cause genetic variations in species... on the other hand, changes in environmental conditions can directly affect the synthesis of active ingredients..." remove repetition and concise the sentence.
Line 63: "Identifying the quality discrepancy among golden camellia from different growing areas is crucial to the development of the golden camellia industry." Change to "Assessing quality differences among golden camellia from various regions is essential for advancing the industry."
Line 93: "Of the 14 planting bases, except two ones located in Hepu County in Guangxi (with a geographical distance of 19 km), all have a geographical distance greater than 50 km from each other." "Two ones" is awkward, and "geographical distance" is repetitive.
Line 95: "Plant individuals from each planting bases formed a natural population, and 14 natural populations were included in this study." "Plant individuals" is awkward and unclear; "each planting bases" is grammatically incorrect.
Lines 101-102: "All individuals of C. nitidissima var. phaeopubisperma at the 14 plant bases are originated from Fangchenggang, Guanggxi, growing from cuttings taken from wild plants in Fangchenggang for more than 15 years." "Are originated" is incorrect
Line 151: "The statistics of genetic diversity and heterozygosity for the 8 SSR loci were shown in Table 3." Use of "were shown" is incorrect for present description.
Line 237-238: "an understanding of the genetic diversity of a species not only can provide the scientific basis for its conservation and utilization, but is of great importance for genetic improvement and innovation of germplasm resources" Awkward expression.
Line 251-252: "indicating the deviation from Hardy-Weingerg equilibrium in most loci of the daughter populations" change to "indicating deviation from Hardy-Weinberg equilibrium in most loci of the progeny populations."
Line 259: "A low heterozygosity across the whole species causes it to be at high risk of extinction" change to "Low heterozygosity across a species increases its vulnerability to extinction."
Line 280-281: "suggesting that population clustering was independent on geographical distances". Grammar issue with "independent on".
Line 288-289: "the total genetic variation of C. nitidissima var. phaeopubisperma was mostly attributed to within individuals, inconsistent with the result of Li et al.(2019)." Unclear expression and spacing issue in reference.
Line 318-319: "So far, genetic variations in C. nitidissima var. phaeopubisperma was independent upon changes in environmental conditions of habitats." "independent upon" is incorrect; should be "independent of."

Experimental design

Experimental design needs clarifications regarding, criteria for selecting 14 species and environmental conditions at each sampling site (e.g., soil composition, climate parameters) as mentioned in basic reporting.

Validity of the findings

The validity of the findings hinges on the inclusion of detailed information regarding the experimental design, as recommended in the basic reporting section.

Reviewer 2 ·

Basic reporting

The manuscript entitled "SSR Marker-Based Genetic Diversity and Structure Analyses of Camellia nitidissima var. phaeopubisperma from Different Populations" uses SSR markers to assess the genetic diversity and population structure of Camellia nitidissima var. phaeopubisperma across various populations. The study aimed to understand genetic variation and differentiation among populations, providing valuable insights for the conservation and management of this species. It could be a significant addition to the journal. However, following queries need to be addressed before further processing.
1. There is inconsistency in the spellings of Camellia genus throughout the manuscript. Ensure the consistency and correctness of spellings.
2. The experimental work focuses on a single variety, Camellia nitidissima var. phaeopubisperma, rather than the species Camellia nitidissima. It is recommended to consistently refer to the study as examining the 'variety' rather than the 'species' throughout the manuscript.
3. In Abstract section, the numbers in brackets (0.25 0.5) are ambiguous. Clarify it.
4. Improve the grammar and sentence structure at Line 52 & 58.
5. Ensure proper formatting of the botanical names according to the rules of nomenclature. Also recheck the formatting of in-text citations.
6. Line 69 & 72, authors may focus on plants only, instead of mentioning about animals along with.
7. The content of line 84 is more appropriate for methodology section.
8. Line 87, there is a need to clearly state the objectives, instead of predicting results.
9. Line 93, "except two ones" means?
10. Line 139, the phrase “is obtained” should be placed outside the bracket for proper grammar.
11. Line 209, the number 17 appears to be repeated. Clarify it.
12. Line 243, there is an ambiguity in the use of words cities and countries, as both are not same.
13. Line 295, the information looks repeating several times throughout the manuscript. Moreover, the introduction about species and variety should be included in the introduction, not later in the manuscript.
14. Figure captions should be appeared below the figures.
15. The conclusion could be elaborated a little, with special focus on major findings, implications, future recommendations and concluding statement.

Experimental design

The experimental design is appropriate to effectively assess the genetic diversity and population structure in Camellia nitidissima var. phaeopubisperma.

Validity of the findings

The findings are valid and supported by proper data collection and statistical analysis, providing valuable insights into genetic variation across different populations of the variety.

Reviewer 3 ·

Basic reporting

yes

Experimental design

yes

Validity of the findings

ok

Additional comments

Review peerj-108071

There were many published papers discussing SSR marker-based genetic diversity. Therefore, the authors need to highlight the novelty of this report. The transcriptomics alone faces challenges as biological organism might develop adaptive responses which finally can not only be explained from transcriptomics analysis. This should be discussed further in the section perspective. Integrated metabolomics might be helpful for further explanation. The authors can search database like Web of Science with integrated metabolomics (Title) AND adaptive response (Title) get reference to discuss it further.

For genetic analysis, matrix effect of the Camellia nitidissima is a key factor deserves further discussion. In this manuscript, it seems this matrix effect was negligible. However, the reasons causing matrix compounds removal for good assay accuracy could be discussed. The authors can search database like Web of Science with matrix compounds removal (Title) AND assay (Title) to get reference to improve the discussion.

Figures/Tables: add data points should be presented with statistical results. For example, the standard deviations or similar parameters should be added and statistical analysis among the groups should be performed and labelled.

---

## Round 0.2 · Minor Revisions

Dear Authors

The manuscript needs a very minor final revision before publication. The authors are invited to revise the paper considering all the suggestions made by reviewer 2.

With Thanks

Reviewer 1 ·

Basic reporting

I have reviewed the revised manuscript and am pleased to see that the authors have addressed most of the concerns raised during the initial review. The revisions have improved the clarity. The study provides valuable insights into the genetic diversity of Camellia nitidissima var. phaeopubisperma. With most of the revisions satisfactorily incorporated, the manuscript can now be accepted for publication.

Experimental design

Experimental design is sufficiently detailed.

Validity of the findings

Results seems promoising.

Reviewer 2 ·

Basic reporting

The revised version of manuscript entitled "SSR marker-based genetic diversity and structure analyses of Camellia nitidissima var. phaeopubisperma from different populations" has been reviewed, along with the authors' responses to reviewers' comments.
Authors have addressed all the concerns raised by the reviewers, to improve the clarity and quality of the manuscript. The overall presentation and organization of the manuscript now meets the scientific and ethical standards required for publication in a prestigious journal. The revised version of the manuscript can now further be processed.

Following typing/formatting errors need to be checked for the final version of manuscript:
There is a typing error at line 103 "being far way from each other".
At few places, there is a need to remove extra spaces.
In reference section, some of the scientific names need italicization.
Line 497, un-italicize word "spp".

Experimental design

The authors have clearly described the experimental procedures, ensuring reproducibility and reliability of the results. The revisions made for this section further clarify the key methodological aspects.

Validity of the findings

The authors have addressed all the concerns raised about the presentation and discussion of findings and conclusion section as well.

Reviewer 3 ·

Basic reporting

the revised version meets the requirement

Experimental design

the revised version meets the requirement

Validity of the findings

the revised version meets the requirement

Additional comments

The authors have addressed the questions quite well. The quality of the manuscript has been improved significantly. There are no further comments and the current version is acceptable for publication.

---

## Round 0.3 · accepted · Accept

Dear Authors,

I am pleased to inform you that the manuscript has improved after the last revision and can be accepted for publication.

Congratulations on accepting your manuscript, and thank you for your interest in submitting your work to PeerJ.

With Thanks